# Factors Contributing to Burnout among Healthcare Workers during COVID-19 in Sabah (East Malaysia)

**DOI:** 10.3390/healthcare10061068

**Published:** 2022-06-09

**Authors:** Nicholas Tze Ping Pang, Noor Melissa Nor Hadi, Mohd Iqbal Mohaini, Assis Kamu, Chong Mun Ho, Eugene Boon Yau Koh, Jiann Lin Loo, Debbie Quah Lye Theng, Walton Wider

**Affiliations:** 1Faculty of Medicine and Health Sciences, Universiti Malaysia Sabah, Kota Kinabalu 88400, Sabah, Malaysia; nicholas@ums.edu.my (N.T.P.P.); assis@ums.edu.my (A.K.); cmho@ums.edu.my (C.M.H.); jiannlinloo@gmail.com (J.L.L.); bm17110083@student.ums.edu.my (D.Q.L.T.); 2Department of Psychiatry and Mental Health, Hospital Tunku Fauziah, Kangar 01000, Perlis, Malaysia; 3Department of Psychiatry and Mental Health, Hospital Tawau, Tawau 91007, Sabah, Malaysia; iqbalmohaini_85@yahoo.com; 4Department of Psychiatry, Universiti Putra Malaysia, Serdang 43400, Selangor, Malaysia; eugene@upm.edu.my; 5Faculty of Business and Communication, INTI International University, Nilai 71800, Negeri Sembilan, Malaysia

**Keywords:** burnout, coping, COVID-19, healthcare workers, perceived social support

## Abstract

The third wave of COVID-19 in Malaysia has significantly strained the healthcare system of the country and increased the level of burnout among the healthcare workers (HCWs) in the country. Therefore, this study aimed to identify the various factors associated with burnout among HCWs. A cross-sectional study was conducted among 150 HWCs in Kota Kinabalu, Sabah, Malaysia. An online survey was administered using the Copenhagen Burnout Inventory, Multidimensional Scale of Perceived Social Support, Brief COPE, and Fear of COVID-19 scales. Pearson correlations were assessed amongst all variables. Subsequently, a multiple linear regression analysis was performed using burnout dimensions as dependent variables. Multiple linear regression results showed: (a) lower work-related burnout (β = −0.217, *p* < 0.01) among married HCWs; (b) higher personal-related burnout (β = 0.228, *p* < 0.01), work-related burnout (β = 0.425, *p* < 0.01), and client-related burnout (β = 0.359, *p* < 0.01) among doctors; (c) fear towards COVID-19 was significantly associated with client-related burnout (β = 0.243, *p* < 0.01); (d) an avoidant coping strategy was significantly associated with personal-related burnout (β = 0.322, *p* < 0.01); (e) social support from family was significantly associated with personal-related burnout (β = −0.264, *p* < 0.01), work-related burnout (β = −0.186, *p* < 0.05), and client-related burnout (β = −0.326, *p* < 0.01);(f) and social support from friends was significantly associated with work-related burnout (β = −0.202, *p* < 0.05). This study demonstrated significant theoretical contributions and clinical implications in the healthcare system in Sabah by addressing the impact of various factors on burnout among HWCs.

## 1. Introduction

The COVID-19 pandemic has caused great upheaval in the world, causing multiple physical, psychological, and economic sequelae [1,2,3]. In Malaysia, the pandemic was fairly well-controlled since its first outbreak in January 2020. The implementation of the Movement Control Order (MCO) during the second wave of outbreak in March 2020, with gradual relaxation of the standard operating procedures (SOP) as Malaysia moved forward into Conditional MCO (CMCO) and subsequently Recovery MCO (RMCO), played a role in flattening the epidemiological curve of COVID-19 [4,5]. Interstate domestic travel including across the South China Sea barrier in between East and West Malaysia was gradually allowed during RMCO, which was implemented in June 2020 and extended until the end of December 2020. An avertable trigger in September 2020, however, made a third wave of COVID-19 in Malaysia inevitable. The start of the third outbreak caused the national healthcare systems to collapse [5,6]. Poor interstate observation and implementation of SOPs contributed to the rapid transmission of the disease, hence increasing number of COVID-19 cases in Malaysia, particularly in Sabah [6].

### Literature Review

Healthcare workers in Sabah were inundated as the situation worsened precipitously, and deployment from other parts of Malaysia was abruptly required to help support the existing staff in Sabah. Nevertheless, this very sudden surge of COVID-19 cases and abrupt escalation in duty hours, duration, and volume contributed to the emergence of potential burnout syndromes in healthcare workers in Sabah [4,7,8]. On top of that, the sudden implementation of MCO further restricted interstate as well as interdistrict travel, hence cutting off the psychosocial support of healthcare workers whose family were out of physical reach during this period of difficult time. As most of the Malaysian doctors working in Sabah were originally from West Malaysia, frequent travel back and forth to their hometown had been a norm prior to the pandemic. The relaxation of travel SOPs during the RMCO period shed some light for a short while before the third wave began. The reimplementation of the two-week quarantine policies for travel into and out of the Sabah state, together with the increasing workforce demands, prohibited the idea of reuniting with their loved ones. The Mental Health and Psychosocial Support Service (MHPSS) was launched from the beginning to help support and monitor the mental wellbeing of the staff [9].

Burnout is defined as a psychosocial syndrome involving feelings of emotional exhaustion, depersonalization, and diminished personal accomplishment at work [10]. Emotional exhaustion is a situation where, owing to a lack of energy, workers perceive they are no longer able to participate on an emotional level [11,12]. Depersonalization entails the development of negative attitudes and feelings towards people for whom work is done, to the point where they are blamed for the subject’s own problems [12]. Diminished personal accomplishment is a tendency in professionals to negatively value their own capacity to carry out tasks and their capacity to interact with the people for whom the tasks are performed and then to feel unhappy or dissatisfied with the results [10,13]. There are various factors leading to burnout that have been studied over the years, both in relation to the pandemic globally and also in a small group of anesthetists in the Malaysian setting [12,14,15,16,17]. In a globally conducted study with participation of 60 countries, it was reported that fifty-one percent of healthcare workers worldwide experienced burnout during the COVID-19 pandemic [12]. In Malaysia itself, a mixed-method study was conducted that concluded that more than half of Malaysian healthcare workers experienced burnout during the COVID-19 pandemic [14]. Therefore, a study is warranted to determine the associated factors that contributed to burnout among Sabahan healthcare workers during the COVID-19 period.

Taking into account the conservation of resources theory (COR) [18], this study postulated that sociodemographic, social, and personal resources can influence burnout. Through effective coping skills and a good support system, individuals are better able to cope with burnout in their job [19]. The COR theory has been used by many studies to demonstrate the influence of safety on psychological stress [20]. In addition, the COR theory also was used to determine the influence of social support on burnout [21]. In spite of stressful life situations, people who lack essential personal, social, and material resources are more vulnerable to experience distress [22]. This study aimed to document the various factors contributing to burnout among the healthcare workers in Sabah during the third wave of the COVID-19 pandemic. The findings may potentially aid in improving support and facilities as part of supporting staff wellbeing by designing targeted measures that can be better tailored to specific risk factors rather than using universal primary prevention measures at greater cost. We focused on Sabah as it was the state hit with the worst pandemic situation during the third wave of COVID-19.

Hence, the specific objectives were as follows: (1) to measure the fear towards COVID-19 among healthcare workers; (2) to identify burnout among the healthcare workers involved in the pandemic onset and management; (3) to identify whether coping skills, perceived support system, and other demographic variables may associated with burnout.

## 2. Materials and Methods

### 2.1. Participants

This was a cross-sectional study analyzing burnout and the associated factors among healthcare workers in Sabah during the COVID-19 pandemic. The target population was all healthcare workers in Sabah.

### 2.2. Data Collection

Data were collected with homogeneous convenient sampling approach using an online survey administered via the internet, through e-mail, and via WhatsApp from 1 December 2020–30 April 2021. The data were collected over a long period of time as it was difficult to achieve the required sample size in a short duration due to high workload. The Sabah state remained in the same level of heightened caseloads throughout that period. These collection methods adhered to COVID-19 social distancing precautions for cross-infection by reducing direct contact and contamination. Participants were not required to have a Google account or sign in to answer the survey. The form was designed to allow only single submission and could not be amended once submitted. The inclusion criterion was all healthcare workers involved with the COVID-19 management. The exclusion criterion was incomplete data. A total of 150 responses were received from respondents, of which all responses were usable. We obtained a response of 100% of the participants without specifying the number of the target healthcare population present in Sabah during the period of administration of the questionnaire. There is no rule of thumb that applies to the sample size needed for a study depends on many factors; however, the general rule of thumb is no less than 50 participants (VanVoorhis & Morgan, 2007).

### 2.3. Instruments

The instruments included in the forms were all self-administered instruments and were as follows:

#### 2.3.1. Copenhagen Burnout Inventory (CBI)

The CBI consists of three sub-scales measuring personal burnout, work-related burnout, and client-related burnout for use in different domains [23]. All three sub-scales were rated using a five-point Likert scale. The scales differentiated well between occupational groups in the human service sector, and correlations with other measures of fatigue and psychological well-being can be found [23]. Furthermore, the three sub-scales predicted future sickness absence, sleep problems, use of analgesics, and intention to quit. The Malay version of the CBI had satisfactory psychometric properties, with the overall Cronbach’s alpha being more than 0.7 [24].

#### 2.3.2. Coping Orientation to Problems Experienced Inventory (Brief COPE)

Brief COPE is a self-administered questionnaire with 28 items that measures 3 dimensions of coping mechanisms, namely, problem-oriented coping, emotion-oriented coping, and dysfunctional coping. For each item, respondents answered using a four-point scale, ranging from 1 (“I have not been doing this at all”) to 4 (“I have been doing this a lot”). The Malay version of the Brief COPE was used, which had good internal consistencies ranging from 0.51 to 0.99, with Cronbach’s alpha ranging from 0.25 to 1.00 [25].

#### 2.3.3. Multidimensional Scale of Perceived Social Support (MSPSS)

This is a self-administered, 12-item questionnaire that measures perceived social support from three specific sources (family, friends, and significant others). For each item, respondents answered using a seven-point scale, ranging from 1 (“Very strongly disagree”) to 7 (“Very strongly agree”). The Malay version of the MSPSS was used, which had a good internal consistency (Cronbach α of 0.89) [26,27].

#### 2.3.4. Fear of COVID-19 Scale (FCV-19S)

This is a self-reported questionnaire containing a 7-item scale measuring fear related to COVID-19. It is scored on a five-point Likert response ranging from 1 (strongly disagree) to 5 (strongly agree), with possible scores ranging from 7 to 35. Higher scores indicate more severe fears of COVID-19. The Cronbach α value for the Malay version of the FCV-19S was 0.893, indicating very good internal reliability [28].

#### 2.3.5. Demographic Data

This was a pro forma form that collected data regarding the demographic particulars of the subjects being studied (age group, gender, marital status, involvement with COVID-19, and healthcare worker category).

### 2.4. Ethical Consideration

The study was conducted in compliance with the ethical principles outlined in the Declaration of Helsinki and the Malaysian Good Clinical Practice (GCP) Guideline. Ethical approval was obtained from the National Medical Research and Ethics Committee (Approval Code: NMRR-20-2494-57139).

### 2.5. Data Analysis

All data were analyzed using the Statistical Package for the Social Sciences (SPSS), Version 26.0. Descriptive statistics assessed the demographic variables using frequency analysis, whereas the Mann–Whitney U test was utilized to determine the significant association between demographic variables and fear of COVID-19. Pearson correlations were calculated between all study variables. Multiple linear regression was subsequently utilized to test which factors were related to burnout in the Sabahan healthcare workers, with the level of significance set as *p* < 0.05. In this study, age, gender, marital status, involvement with COVID-19, and healthcare worker category were set as dummy variables to ensure accuracy.

## 3. Results

### 3.1. Descriptive Statistics

Table 1 demonstrates that the majority of participants were female (78.0%), less than 40 years old (89.3%), involved as frontliners in COVID-19 (83.3%), and single (52.0%). Table 1 also suggests that most respondents only suffered from lower levels of client- and work-related burnout, at 60.0% and 72.0%, respectively. However, the majority of respondents suffered from moderate levels of personal-related burnout, at 44.0%.

### 3.2. The Differences of Fear of COVID-19 across Demographic Variables

As demonstrated in Table 2, the Mann–Whitney U test indicated that there was no significant difference in fear toward COVID-19 between age groups, gender, marital status, and involvement with COVID-19. However, the healthcare worker category had a significant association with fear of COVID-19. Specifically, doctors reported having a significantly higher fear of COVID-19 than other healthcare workers.

### 3.3. Pearson Correlation Analysis

As per Table 3, perceived social support from a significant other, perceived social support from family, and perceived social support from friends are negatively correlated with all dimensions of burnout, whereas the healthcare worker category (doctor) is positively correlated with all dimensions of burnout. Avoidance coping is positively correlated with personal-related burnout and work-related burnout. Problem-focused coping is negatively correlated with work-related burnout and client-related burnout. Additionally, fear of COVID-19 is positively correlated with client-related burnout. Meanwhile, marital status is negatively correlated with personal-related burnout and work-related burnout. Other variables such as age, gender, emotion-focused coping, and involvement with COVID-19 were not correlated with burnout variables. Therefore, the linear multiple regression was further performed to determine the association between healthcare worker category (doctor), marital status (married), fear of COVID-19, problem-focused coping, avoidance coping, perceived social support from significant other, perceived social support from family, and perceived social support from friends and all dimensions of burnout.

### 3.4. Multiple Regression Analysis

Table 4 shows the results of the linear multiple regression. Married healthcare workers had significantly lower work-related burnout (β = −0.217, *p* < 0.01) compared with single healthcare workers. The doctors had significantly higher personal-related burnout (β = 0.228, *p* < 0.01), work-related burnout (β = 0.425, *p* < 0.05), and client-related burnout (β = 0.359, *p* < 0.05) compared with other healthcare worker categories. The fear towards COVID-19 was significantly associated with client-related burnout (β = 0.243, *p* < 0.05) among the healthcare workers. The avoidant coping strategy was significantly associated with personal-related burnout (β = 0.322, *p* < 0.05), while social support from friends was significantly associated with work-related burnout (β = −0.202, *p* < 0.05). Last but not least, social support from family was significantly associated with personal-related burnout (β = −0.264, *p* < 0.01), work-related burnout (β = −0.186, *p* < 0.05), and client-related burnout (β = −0.326, *p* < 0.01).

## 4. Discussion

The purpose of this study was to determine the association between the age group, involvement with COVID-19, healthcare worker category, marital status, perceived support system, coping styles, and fear of COVID-19 with burnout among HCWs in Sabah. Our findings revealed that doctors had a significantly higher fear of COVID-19 and burnout in all dimensions compared with other healthcare worker categories. Perceived social support from family served as buffer against all burnout dimensions. Meanwhile, married healthcare workers and those who received social support from friends showed lower work-related burnout. Avoidance coping resulted in higher personal-related burnout. Finally, fear of COVID-19 could lead to significant burnout related to clients.

This study clearly demonstrated that doctors were at higher risk of burnout in all categories measured within the construct, compared with other categories of healthcare workers. This finding correlates with previous findings that demonstrate high levels of psychopathology in healthcare workers in a similar Malaysian setting, but this study is crucial in that it explores other factors associated with burnout [29]. Comparatively, high levels of personal-related, work-related, and patient-related burnout were also found in a study conducted among doctors in Sabah even before the pandemic, with rates of 57.1%, 48.8%, and 30.4%, respectively [30]. Due to the higher transmission rate of COVID-19 viruses, it is commonly reported that doctors experienced fear of becoming infected and transmitting it to their immediate family members [31,32]. Furthermore, individual factors such as sociodemographic and coping styles were instrumental in influencing burnout. For instance, an avoidant coping strategy is associated with higher personal-related burnout, whereas being married contributed to lower work-related burnout. These findings correlate with a number of studies conducted that focused on the impact of coping skills and family support on the psychological health of healthcare professionals [33,34].

Considering work-related burnout, it is interesting that social support from families and friends were the two significant factors that remain aligned with a previous study [17]. This suggests how crucial it is for policymakers not to separate frontline healthcare workers and their friends or families, as they can be a crucial psychological intervention with far better efficacy than pharmacological or psychological therapy. The Sabah state is on Borneo island, accessible only by plane, and consequently enjoys more autonomy over its immigration and infectious disease quarantine policies [35]. Renewed lockdowns have meant that healthcare workers, especially those who are working in the Sabah state, which was the target of this study, faced extreme difficulties in frequently seeing their families due to the two-week quarantines required upon exiting and entering Sabah state. Due to a chronic shortage of doctors in the Sabah state, the vast majority of higher-level doctors at a specialist level are from West Malaysia, who then faced the prospect of not seeing their family for months on end as they were trapped behind a commercial flight and immigration near embargo. Hence, more doctors opted either to apply for permanent transfers back to West Malaysia rather than continue serving in Sabah with regular trips back or to quit government service, exacerbating the burnout for the doctors who remained in government service in Sabah. Hence, those who were not married may have resorted to more avoidant coping strategies, including alcohol use and disengagement, as they were unable to turn to their family for their problem- or emotion-oriented coping styles [36].

Another important factor to address is the fear of COVID-19, which is more relevant outside the work sphere for Sabahan frontline healthcare workers; this finding has been replicated in Malaysian and Indonesian settings [28,37,38]. This suggests that healthcare workers felt reassured at work as they were using personal protective equipment and also enjoyed the benefits of all patients at work having already been tested for COVID-19 before arrival, paradoxically making work a safer place to be when compared with outside work [39]. However, this phenomenon may not extend outside the hospital-based environment; hence, it would be of value for psychological interventions combating fear of COVID-19 to target certain skills, e.g., acceptance of the risk of COVID-19 and dispelling unhelpful thoughts about disease contraction outside work [9]. Cognitive behavioral therapy skills, including Socratic questioning and questioning logical fallacies, may actually be helpful as well for such healthcare workers [40], as the vast majority of clusters in Malaysia have been occupational, rather than from random contact with outside strangers [4].

### 4.1. Limitations

There are a few limitations in this study that need to be addressed. Firstly, the convenient sampling used for data collection increased the chance of bias, in which the random nature of the sample selection may not represent the target population accurately. Secondly, the sample was taken with a special interest in healthcare workers in Sabah during the period of third wave of COVID-19; therefore, the results are specific to the involved area and period of time and may not necessarily be generalized to other regions affected by the pandemic. Lastly, the relatively long period of 5 months for sample collection poses its own variability in the experience of burnout of healthcare workers at different point of time within the said period, taking into account ongoing policy changes to the COVID-19 outbreak response measures by the government, which may have given different level of aids to our healthcare in battling the pandemic situation.

### 4.2. Implications

With the help of the COR theory, our study provided support to previous studies and helped to examine the possible factors influencing burnout during COVID-19 in the context of HCWs in East Malaysia (Sabah). The findings of this study will add value to the COR theory and healthcare literatures. This study looked at the burnout of HCWs, and it provided insight into an area of improvement so that other researchers can consider the sociodemographic, social, and personal resources of HCWs as part the cause. Several practical implications also emerged. To deal with issue of burnout among HCWs, policymakers need to take into account the demographic and psychological traits and the support systems of the HCWs. The findings of this research indicate that doctors have a higher rate of burnout when compared with other types of healthcare workers. Hence, policymakers need to make appropriate efforts to ensure the sustainability of Sabah’s healthcare service as part of the pandemic crisis response by providing sufficient social support, which includes instrumental support and emotional support to the HCWs, especially doctors.

## 5. Conclusions

This study clearly suggests that Sabahan frontline workers had various factors that influenced burnout and that doctors were disproportionately affected. As the training of doctors incurs a high cost both to the individual and to society, due to the burden of public funding’s subsidizing public medical education, losing doctors as a sequelae of various sources of burnout is a great overall societal loss and can be averted by providing earlier psychosocial support to the extreme emotional and social stressors that COVID-19 work can pose. This study strongly suggests that interventions targeting burnout should be considered by higher authorities so that frontline HCWs receive an appropriate level of support that can assist them in guaranteeing the sustainability of the healthcare system in Sabah. Nevertheless, the generalizability of the data to other parts of the country may be limited due to the nature of homogeneous convenient sampling, as mentioned above, and the difference in the landscape of the healthcare system compared with other parts of the country, suggesting that replicative studies are warranted.

## Figures and Tables

**Table 1 healthcare-10-01068-t001:** Descriptive analysis of the healthcare workers (HCW) enrolled in the study (n = 150).

Variables		N (%)
Age group	Less than 40 years	134 (89.3%)
	40 years and above	16 (10.7%)
Gender	Male	33 (22.0%)
	Female	117 (78.0%)
Marital status	Single	78 (52.0%)
	Married	72 (48.0%)
Involvement with COVID-19 (i.e., frontliner)	No	25 (16.7%)
	Yes	125 (83.3%)
	Doctor	69 (46.0%)
Healthcare worker category	Nurse	57 (38.0%)
	Others	24 (16.0%)
Personal-related burnout	Low	49 (32.7%)
	Moderate	66 (44.0%)
	High	26 (17.3%)
	Severe	9 (6.0%)
Work-related burnout	Low	90 (60.0%)
	Moderate	35 (23.3%)
	High	15 (10.0%)
	Severe	10 (6.7%)
Client-related burnout	Low	108 (72.0%)
	Moderate	27 (18.0%)
	High	12 (8.0%)
	Severe	3 (2.0%)

**Table 2 healthcare-10-01068-t002:** Fear toward COVID-19 based on the demographic characteristics of the healthcare workers (HCWs) enrolled in the study.

Variables	Fear of COVID-19			
	Mean Rank	IQR	Z	Asimp. Sig. (2-Tailed)
Age group				
Less than 40 years	73.87	9899.00	−1.33	0.184
40 years and above	89.13	1426.00		
Gender				
Male	65.03	2146.00	−1.57	0.116
Female	78.45	9179.00		
Marital status				
Single	75.56	5894.00	−0.02	0.985
Married	75.43	5431.00		
Involvement with COVID-19 (i.e., frontliner)				
No	87.92	2198.00	−1.57	0.117
Yes	73.02	9127.00		
Healthcare worker category				
Doctors	84.97	6882.50	−2.90	<0.001
Others	64.38	4442.50		

**Table 3 healthcare-10-01068-t003:** Pearson correlations between variables.

	Variables	1	2	3	4	5	6	7	8	9	10	11	12	13	14	15
1	PFC	1														
2	EFC	0.797 **	1													
3	AC	0.270 **	0.411 **	1												
4	Family	0.281 **	0.208 *	−0.156	1											
5	Friends	0.304 **	0.239 **	−0.092	0.713 **	1										
6	Other	0.271 **	0.290 **	−0.020	0.775 **	0.662 **	1									
7	FOC-19	0.038	0.086	0.155	−0.115	−0.239 **	−0.132	1								
8	Age	0.049	0.050	0.093	−0.190 *	−0.207 *	−0.074	0.119	1							
9	Male	0.019	0.055	−0.162 *	0.101	0.105	0.008	0.104	−0.077	1						
10	Married	0.040	−0.014	−0.044	0.085	−0.055	0.153	−0.001	0.360 **	−0.100	1					
11	Frontliner	−0.116	−0.169 *	−0.069	−0.076	−0.079	−0.010	−0.117	0.155	0.108	0.036	1				
12	Doctor	−0.236 **	−0.031	0.083	−0.054	−0.031	−0.038	−0.220 **	−0.232 **	−0.220 **	−0.084	0.018	1			
13	PRB	−0.069	0.131	0.382 **	−0.326 **	−0.255 **	−0.192 *	0.125	0.064	0.017	−0.165 *	0.045	0.269 **	1		
14	WRB	−0.214 **	−0.029	0.205 *	−0.371 **	−0.335 **	−0.317 **	0.078	−0.027	−0.037	−0.257 **	0.075	0.459 **	0.724 **	1	
15	CRB	−0.263 **	−0.095	0.116	−0.374 **	−0.368 **	−0.326 **	0.201 *	−0.072	0.027	−0.126	−0.050	323 **	0.602 **	0.730 **	1
	Mean	2.89	2.67	2.00	5.22	5.01	5.05	15.56	0.780	0.480	0.833	0.460	0.197	1.63	1.40	0.460
	SD	0.686	0.573	0.511	1.48	1.57	1.77	5.82	0.310	0.415	0.501	0.374	0.862	0.915	0.723	0.500

Notes. PFC: problem-focused coping; EFC: emotion-focused coping; AF: avoidance coping; Family: perceived social support from family; Friends: perceived social support from friends; Other: perceived social support from significant other; FOC-19: fear of COVID-19; Age: 40 years and above; Frontliners: involvement with COVID-19; Doctor: healthcare worker category; PRB: personal-related burnout; WRB: work-related burnout; CRB: client-related burnout. * *p* < 0.05; ** *p* < 0.01.

**Table 4 healthcare-10-01068-t004:** Factors affecting the burnout among healthcare workers (HCW).

Dependent Variables	Independent Variables	Standardized β	t	*p*	Adjusted R^2^	*F*	*p*
Personal-related burnout	Avoidance coping	0.322	4.482	0.00	0.255	18.04	<0.01
	Perceived social support from family	−0.264	−3.681	0.00			
	Healthcare worker category (doctor)	0.228	3.216	0.00			
Work-related burnout	Healthcare worker category (doctor)	0.425	6.499	0.00	0.370	22.89	<0.01
	Perceived social support from family	−0.186	−1.968	0.05			
	Marital status (married)	−0.217	−3.274	0.00			
	Perceived social support from friends	−0.202	−2.143	0.03			
Client-related burnout	Perceived social support from family	−0.326	−4.621	0.00	0.272	19.55	<0.01
	Healthcare worker category (doctor)	0.359	4.989	0.00			
	Fear of COVID-19	0.243	3.359	0.00			

## Data Availability

Data are available from the authors upon reasonable request.

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
