# Peer review of "Factors Contributing to Burnout among Healthcare Workers during COVID-19 in Sabah (East Malaysia)"

_healthcare, 2022, doi:10.3390/healthcare10061068_

Round 1

Reviewer 1 Report

This article estimates the factors contributing to burnout among Healthcare Workers during COVID-19 in Sabah. The topic seems to be very interesting in conducting feelings of depression, anxiety, traumatic stress, and burnout among Healthcare Workers during pandemic era. The introduction is clear and concise and gives a good rationale for the strong emotional impact of the epidemic and create health situations which possible can lead to long-term psychiatric complications in this special population. All the figures and tables are easily readable, correct, and informative. Results and discussion present scientific soundness and interest to the readers. In my view, after carefully reading the article, although it was a cross- sectional online survey involving an auto-selected small sample, I think the article can be approved for publication. 

Author Response

Dear examiner,

We are grateful for your consideration of this manuscript. Thank you.

Reviewer 2 Report

In the M&M it is specified that the sample is homogeneous and convenient but it would be appropriate to specify through which platforms the participants were reached and with criterion classrooms they were chosen (facebook, mailinglist, website ...) and if the link was potentially sent to the whole target population, also specifies that we obtained a response of 100% of the participants without however specifying the number of the target healthcare population present in Sabah during the period of administration of the questionnaire.
In the results, Table 3 is hard to read and the significant results are hardly highlighted and legible.

Author Response

Dear Examiner,

We are grateful for your consideration of this manuscript, and we also very much appreciate your suggestions, which have been very helpful in improving the manuscript. All the comments we received on this manuscript have been taken into account in improving the quality.

Reviewer 3 Report

Dear Authors,

I congratulate you all for wiring this interesting and much needed piece of research. Following are my comments/recommendations which might help you in improving the quality of your research paper.

1. In the fourth line of Abstract, I presume that the word “study” is missing after the word “cross-sectional”.

2. Very well written introduction and comprehensive literature review.

3. Please explain about the population and selection of sample size. You have selected a sample of 150 respondents on which basis? How do you believe that this sample is representative of the population you targeted?

4. 2.3.1. Demographic data (This heading should ideally come after the fear of COVID-19 scale. This is my recommendation. You may disagree with this.

5. You may do CFA for your proposed model to establish the validity of the model.

6. My main suggestion to the authors is to use an overarching theory for your proposed model. The most promising theory for your model seems to me could be the Theory of Planned Behaviors (TPB, Ajzen, 1985; 1991). Or you can check the Conservation of Resource Theory (COR; Hobfoll, 1989) if it supports your model.

Cheers

Reviewer

Author Response

(The authors gave the same response as above.)

Reviewer 4 Report

Dear Authors, 

Your work is timely and important.

I have some suggestions that can hopefully help improve your work.

1. You should add a section "Literature Review" after the introduction. You should take the text already produced, from line 63 to line 95 and insert it in the new Literature Review section.

2. You should also develop the literature review more, in order to have a more complete and consistent review of the state-of-the-art. You can add more studies, as the literature in this topic is vastly covered. You can take a look at this study: 10.3390/ijerph18052425

3. In the Discussion section you should add a subsection called "implications", where you discuss the theoretical and practical implications of your study. You can also add suggestions for policy makers. 

4. In the Conclusions section you should mention the future roads of research.

Good work!

Author Response

(The authors gave the same response as above.)

Reviewer 5 Report

Dear Authors,

Thank you for conducting this study. Your effort is highly appreciated. 

I have some comments:

Regarding the phrasing, please find attached the file where a have highlighted areas that need to be rephrased.

And please exclude comments that could have political implications, unless you can find a reliable source to cite. So, please exclude "Precipitated by the state election" and anything related to that comment throughout the text.

Author Response

Dear Examiner,

We are grateful for your consideration of this manuscript, and we also very much appreciate your suggestions, which have been very helpful in improving the manuscript. All the comments we received on this manuscript have been taken into account in improving the quality.

This manuscript is a resubmission of an earlier submission. The following is a list of the peer review reports and author responses from that submission.

Round 1

Reviewer 1 Report

I am really surprised to see such a quantitative primary survey based manuscript where none of these things addressed:
1) A significant literature review.

2) Sound theoretical modeling-based argumentation in the literature. 

3) External or internal validity testing of the instrument and collected data.

4) Practical and theoretical implications with sound argumentation.

Reviewer 2 Report

The article tackles an interesting topic but has two major flows which cannot be overcome in a simple revision. 1) It lacks a proper literature review (and contains a lot of self-citations) and 2) it offers scarce methodological details. Because I cannot check the validity of the survey items, I decided to reject this paper. Some comments below.

I am confused by the abstract: did the election cause the third wave, or the opposite? In any case, I would avoid making political claims in a scientific paper unless strictly necessary.

Introduction: what is your contribution to the literature? How is Malaysia different from the other cases reported in previous studies?

Materials:

The authors claim "estimated sample size was calculated to range from 154 to "193" but their sample size, according to Table 1, was 150

please describe the Maslach Burnout Inventory. Why is your approach better?

Where does the brief COPE come from? Where does the MSPSS come from? Where does the FCV-19S come from?

If the authors have developed these items themselves, it should be explicitly stated and motivated.

Also, the items of the survey should be explicitly stated, either in the text or in the appendix. I don't have a way to check where the survey items come from, and whether they're a valid measure.